# POLIA: Policy Optimization with Visual-Object-Level Intrinsic Advantage for Multimodal Reasoning

Yiran Zeng [1]   Da Chen [1]   Hangyu Mao [2]   Yuanxing Zhang [2]   Pengfei Wan [2]   Mengchen Zhao [1]

## Abstract

Recent advances in group-based reinforcement learning (RL) greatly improve LLMs' ability in text reasoning. Yet, these methods lack sufficient modeling of multimodal information, leading to significant reasoning hallucination. In this work, we propose POLIA, a novel group-based RL method with visual-object-level intrinsic advantage for multimodal reasoning. POLIA introduces two advantage computation stages over candidate answers and visual objects, respectively. The answer-level extrinsic advantages are computed based on the extrinsic rewards of a group of candidate answers. Moreover, we compute an intrinsic advantage for each visual object based on its confidence score and reference relations with final answers. Intuitively, the intrinsic advantage of an object reflects its potential contribution to the correct answer. This two-stage advantage computation ensures an accurate credit assignment mechanism over multimodal reasoning sequences with multiple visual objects. Experimental results on diverse multimodal reasoning benchmarks show that POLIA significantly outperforms open MLLMs and strong baselines. Code is available at https://github.com/dudu115/POLIAcode.

## 1. Introduction

Reinforcement learning (RL) has been demonstrated as an effective tool for enhancing reasoning in large language models (LLMs) (Zhou et al., 2025; Wang et al., 2025c). For multimodal large language models (MLLMs), reasoning requires synthesizing and analyzing information from different modalities (Yin et al., 2024). Existing RL methods, such as GRPO (Shao et al., 2024b), fail to differentiate between different modalities, resulting in significant reasoning hallucinations in trained models (Zhang et al., 2025a; Tu et al., 2025). For example, the models may ignore the visual information during reasoning, or generate seemingly correct answers that disobey the visual facts (Guo et al., 2025). These issues urge novel RL methods to substantially improve the reasoning ability of MLLMs (Huang et al., 2025; Shen et al., 2025; Yang et al., 2025).

Using RL to improve reasoning in MLLMs faces three main challenges. First, there is no explicit and structured modeling of visual evidence, which is usually vaguely referred to as visual-related information in multimodal reasoning (Fan et al., 2025; Zhou et al., 2025). Without a clear emphasis on visual evidence, MLLMs often struggle to extract the most useful visual information from input images. Second, it is hard to construct effective reward signals to guide the reasoning process (Bai et al., 2024), since existing datasets only provide the final answers as references to judge the model's outputs (Wu et al., 2025b). Consequently, models might be encouraged to achieve fake high reward by reward hacking. Third, existing RL approaches lack an explicit credit assignment mechanism for visual evidence. Without a clear and reasonable credit assignment mechanism, models may easily ignore the key visual evidence that contributes most to the final answer (Zhang et al., 2025c).

Prior works have demonstrated the effectiveness of RL in multimodal reasoning, yet their improvements are still marginal. Early works extend RL for LLMs to MLLMs by simply mixing visual tokens with text tokens (Yang et al., 2025; Huang et al., 2025). These approaches heavily rely on base MLLMs' capability to distinguish key visual evidence, leading to a poor learning efficiency. Recent works refine the reasoning process by encouraging the model to reference visual tokens when generating answers (Zheng et al., 2025b; Zhang et al., 2025d; Cao et al., 2025; Fan et al., 2025; Wang et al., 2025a). Although they improve the usage of visual information during reasoning, the model may conclude with totally wrong references to visual evidence. In other words, due to the lack of reasoning process reward and explicit credit assignment, the model fails to learn how to reason with the visual evidence.

[1]South China University of Technology, Guangzhou, China [2]Kuaishou Technology, Beijing, China. Correspondence to: Mengchen Zhao <zzmc@scut.edu.cn>.

*Proceedings of the 43rd International Conference on Machine Learning*, Seoul, South Korea. PMLR 306, 2026. Copyright 2026 by the author(s).

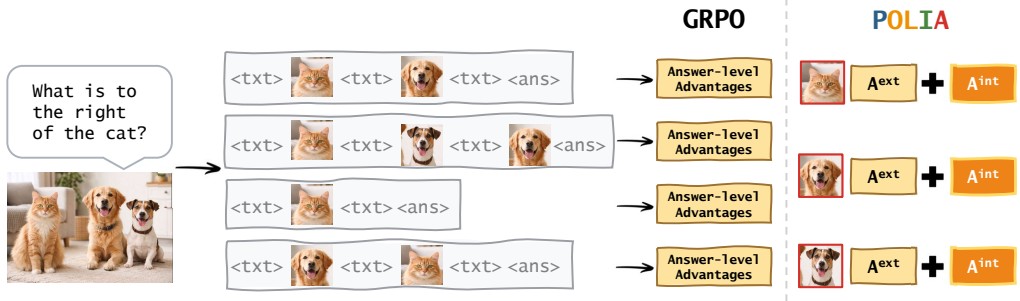

*Figure 1.* Comparison between GRPO and POLIA on multimodal reasoning. **Left**: Traditional GRPO computes advantages only at the answer level. **Right**: POLIA combines answer-level extrinsic advantage ($A^{\text{ext}}$) with visual-object-level intrinsic advantage ($A^{\text{int}}$) for multimodal policy optimization, enabling explicit credit assignment over visual objects and more effective use of visual evidence.

In this paper, we present POLIA, a novel group-based RL method with visual-object-level intrinsic advantage for multimodal reasoning. POLIA extends GRPO by incorporating an additional *intrinsic advantage*, which significantly improves the accuracy of credit assignment on visual evidence. Figure 1 illustrates this comparison. Overall, the advantage computation of POLIA can be divided into two stages. **Firstly**, the candidate answers (including the associated reasoning chains) generated by the MLLM are treated as a large group, where we compute an *extrinsic reward* for each candidate answer by comparing it with the referenced ground-truth answer in the dataset. Hence, we compute an *extrinsic advantage* for each candidate answer following GRPO. **Secondly**, since each candidate answer refers to a unique group of visual objects, we identify several groups of visual objects as visual evidences [1]. In each visual object group, we broadcast the answer-level extrinsic rewards to each visual object with a correction based on its confidence. Then, we compute an intrinsic advantage for each visual object by comparing the corrected object-level rewards within the same visual object group. Intuitively, the intrinsic advantage of an object reflects its potential contribution to the correct answer. *We call it intrinsic advantage because it is computed within a naturally formed visual object group during training.* The above two-stage advantage computation enables a more accurate and fine-grained credit assignment mechanism, which is essential to mitigate hallucination and reward hacking in multimodal reasoning.

Our main contributions are summarized as follows.

- Inspired by human picture reading, we propose a new multimodal reasoning paradigm where the visual evidence is formulated as a group of visual objects. By this way, MLLMs can concentrate on learning the high-level reasoning logic between visual objects.

- We extend GRPO by incorporating a novel intrinsic

---

[1]Note that we formulate the visual evidence as a group of visual objects because humans usually read pictures at the object level.

advantage computation module, which provides fine-grained reward signals on the object level that effectively guide the learning of reasoning process.

- We evaluate POLIA on seven widely used multimodal reasoning datasets, covering complex object counting, spatial relation understanding, mathematical reasoning with visual context, and related tasks. Experimental results show that POLIA significantly improves the reasoning capability of MLLMs.

## 2. Related Works

### 2.1. Reinforcement Learning for LLM Reasoning

RL was adopted early for LLMs through Reinforcement Learning from Human Feedback (RLHF), which guides model updates using human preference signals (Ziegler et al., 2019; Stiennon et al., 2020; Ouyang et al., 2022; Gu et al., 2024). Subsequently, RL methods for enhancing reasoning evolved from PPO (Schulman et al., 2017) and DPO (Rafailov et al., 2023) toward critic-free, group-based algorithms represented by GRPO (Shao et al., 2024b). This formulation samples multiple candidate answers for the same query and estimates advantages within the group, without training a value function, which reduces training cost and facilitates its application to LLMs. Subsequent work, such as Dr. GRPO (Liu et al., 2025a), DAPO (Yu et al., 2025b), GSPO (Zheng et al., 2025a), GMPO (Zhao et al., 2025) and GFPO (Shrivastava et al., 2025), further improves GRPO in various aspects. These methods exhibit a natural fit for LLMs: candidate answers share a common textual context, enabling answer-level rewards to fairly compare reasoning quality. For MLLMs, different answers may rely on different visual evidence, preventing answer-level rewards from fairly comparing reasoning quality. Therefore, adapting RL methods for LLMs to multimodal reasoning requires explicit consideration of visual grounding.

## 2.2. Visual Information Modeling for MLLM Reasoning

Prior work improves multimodal reasoning by introducing more explicit and structured modeling of visual information into the reasoning process. Early efforts mainly rely on prompting strategies to encourage models to incorporate visual information, such as multimodal chain-of-thought formulations (Zhang et al., 2023; Shao et al., 2024a; Li et al., 2025). Subsequent approaches introduce explicit visual evidence representations, enabling models to associate reasoning with corresponding image regions (Peng et al., 2023). More recent methods further enrich visual information modeling by supporting step-by-step interactions with images (Qi et al., 2024). In addition, RL has been applied to learn multimodal reasoning trajectories without intermediate supervision during the reasoning process (Fan et al., 2025; Cao et al., 2025). Despite these advances, existing approaches still struggle to explicitly structure visual evidence into meaningful objects and reason over their relations, making it difficult to learn the high-level reasoning logic needed for accurate multimodal understanding.

## 2.3. Reinforcement Learning for MLLM Reasoning

Building on the success of group-based RL in LLM reasoning, recent work applies critic-free policy optimization to MLLMs (Meng et al., 2025; Shen et al., 2025; Peng et al., 2025; Wang et al., 2025b; Yang et al., 2025). While these methods demonstrate that RL can enhance multimodal reasoning, reward signals defined mainly based on final answers provide limited guidance for the reasoning process and visual information utilization. To address this limitation, some studies refine reward signals by introducing additional constraints on reasoning behaviors (Zhang et al., 2025b; Xia et al., 2025; Xu et al., 2025; Liu et al., 2025b; Wu et al., 2025a). In parallel, other works focus on improving visual reliability by incorporating explicit visual evidence (Wang et al., 2025d; Zhang et al., 2025a; Yu et al., 2025a; Fan et al., 2025; Sarch et al., 2025). However, existing RL methods still rely on final-answer supervision, assigning a single scalar reward to candidate answers that refer to different visual evidence. Therefore, these reward signals fail to provide precise credit assignment over visual evidence, leading to coarse guidance for multimodal reasoning.

## 3. Preliminaries

**Group-based RL.** Group-based RL has been widely used to train LLMs without learning an explicit value function. Given an input $x$, a policy $\pi_{\theta_{\text{old}}}$ samples a group of $N$ candidate answers $\{y_1, \ldots, y_N\}$, each corresponding to a complete reasoning process and final answer. Each candidate answer $y_i$ is assigned a scalar reward $r_i = R(x, y_i)$ based on the overall quality of the generated outcome. Group-based RL computes advantages by comparing rewards within the same group of candidates. Specifically, the advantage of each answer is obtained by normalizing its reward with respect to the group:

$$A_i = \text{GroupNorm}\left(\{r_j\}_{j=1}^N\right).$$

GRPO is a representative group-based RL method that follows this principle by normalizing rewards using the group mean and variance. This critic-free design simplifies training and is well suited for large-scale policy optimization.

**Problem setup.** In multimodal reasoning, each input $x = (I, Q)$ consists of an image $I$ and a query $Q$, and each candidate answer $y_i$ contains both a reasoning chain and a final answer. Different candidate answers may rely on different visual evidence, which we represent as a group of visual objects $S_i = g(x, y_i)$. However, standard group-based RL normalizes rewards across the entire group, implicitly treating all candidates as directly comparable. When $S_i \neq S_j$, such normalization fails to provide precise credit assignment over visual evidence, resulting in coarse guidance for multimodal reasoning. Consequently, group-based RL in multimodal settings requires more precise treatment of visual evidence to support reliable multimodal reasoning.

## 4. Methods

**Motivation.** Existing RL methods for multimodal reasoning optimize rewards at the final-answer level, even though candidate answers may rely on different visual evidence. This limits credit assignment on visual evidence and exacerbates hallucination and reward hacking. We address this issue by computing advantages in two stages: an answer-level *extrinsic advantage* from final-answer supervision, and an object-level *intrinsic advantage* computed within visual object groups of answers referring to the same visual evidence.

**Overview.** As illustrated in Figure 2, we propose POLIA, a policy optimization with visual-object-level intrinsic advantage for multimodal reasoning. POLIA has a two-stage advantage formulation that operates at both the answer level and the visual-object level. By introducing intrinsic advantages over visual objects, POLIA provides more explicit credit assignment on visual evidence. This design enables more reliable optimization under final-answer supervision.

## 4.1. Answer-Level Extrinsic Advantage Computation

Given an image-query input $x = (I, Q)$, we sample a group of $N$ candidate answers from the current policy $\pi_\theta$: $\{c_1, \ldots, c_N\} \sim \pi_\theta(\cdot \mid x)$. We then compute an answer-level extrinsic reward $R(c_i)$ for each candidate answer $c_i$:

$$R(c_i) = \lambda_{\text{ans}} R_{\text{ans}}(c_i) + \lambda_{\text{fmt}} R_{\text{fmt}}(c_i).$$

In practice, $R(c_i)$ is composed of: (i) an answer correctness term $R_{\text{ans}}(c_i)$ that measures the match between the final

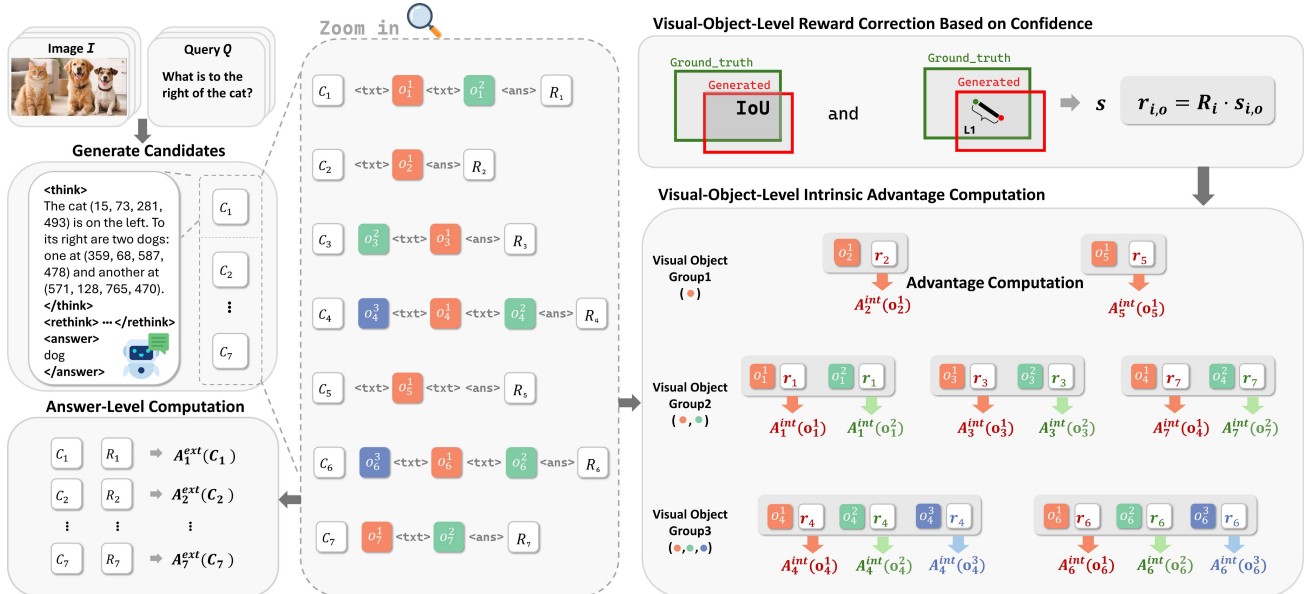

*Figure 2.* Overview of POLIA. Given an image–query input, POLIA samples a group of candidate answers ($C$) and computes answer-level extrinsic advantage ($A^{\text{ext}}$) from the extrinsic reward ($R$). Then, candidates are grouped by the visual objects ($o$) they reference, where the same color indicates the same visual object. For each answer, $R$ is corrected by object confidence ($s$) to obtain $r$. Visual-object-level intrinsic advantage ($A^{\text{int}}$) are computed from $r$ within each visual object group. The policy is optimized by combining $A^{\text{ext}}$ and $A^{\text{int}}$.

answer and the ground-truth answer. It provides the primary supervision signal, which is directly aligned with the evaluation metric; (ii) a format validity term $R_{\text{fmt}}(c_i)$ that enforces the required coordinate format for visual evidence specification in the reasoning chain, which is necessary for subsequent visual object group construction. $\lambda_{\text{ans}}, \lambda_{\text{fmt}}$ balance these components in the overall objective.

Following GRPO, we compute extrinsic advantages $A_i^{\text{ext}}$ by normalizing $R(c_i)$ within the group:

$$A_i^{\text{ext}} = \frac{R(c_i) - \frac{1}{N} \sum_{j=1}^N R(c_j)}{\text{std}(\{R(c_j)\}_{j=1}^N) + \delta},$$

where $\delta$ is a small constant for numerical stability. This groupwise relative comparison preserves the efficiency of critic-free policy optimization and provides a stable answer-level training signal. However, as discussed above, when candidate answers rely on different visual evidence, answer-level rewards become insufficient, motivating explicit and structured visual evidence modeling.

### 4.2. Visual Evidence Modeling

To address the lack of explicit and structured modeling of visual evidence, we represent the visual evidence referenced by each candidate answer as a group of visual objects. This design is motivated by human image interpretation habits and it provides an explicit handle to emphasize and compare visual evidence in multimodal reasoning. Concretely, each candidate answer $c_i$ may contain explicit visual refer-

ences in the form of predicted bounding boxes. We match each predicted box to a ground-truth object box using a predefined matching rule, and denote the matched object identities as an order-invariant set:

$$S_i = \{o_{i,1}, \ldots, o_{i,m_i}\}.$$

We treat $S_i$ as the visual evidence used by $c_i$. Under this formulation, two candidate answers are considered to rely on the same visual evidence only if they induce the same object set. Compared with leaving visual information as vaguely defined visual related information, this representation offers an explicit and structured modeling of visual evidence, and forms the basis for the subsequent visual object group construction under consistent evidence condition.

### 4.3. Visual-Object-Level Intrinsic Advantage Computation

**Visual objects grouping.** Given the visual evidence $S_i$ referenced by each candidate answer $c_i$, we group candidates according to the visual objects they rely on. Specifically, for each visual evidence $S$, we construct a visual object group:

$$\mathcal{G}_S = \{c_i \mid S_i = S\}.$$

Candidate answers within the same group rely on identical visual evidence and are therefore treated as logically comparable under a consistent evidence condition. This grouping provides a structured basis for within-group comparison and subsequent reward propagation.

**Visual-object-level reward correction based on confidence.** For a candidate answer $c_i \in \mathcal{G}_S$, we consider each referenced visual object $o \in S_i$ and compute an object confidence score $s_{i,o} \in [0,1]$ that measures how reliably $c_i$ refers to $o$. In our implementation, $s_{i,o}$ is derived from the matching quality between the predicted bounding box in $c_i$ and the ground-truth box of object $o$, computed as a weighted combination of IoU and normalized $L_1$ distance, followed by clipping. We define the corrected object-level reward of each visual object as:

$$r_{i,o} = R(c_i) \cdot s_{i,o}.$$

**Visual-object-level intrinsic advantage computation.** We compute an intrinsic advantage $A_{i,o}^{\text{int}}$ for each object by comparing the $r_{i,o}$ within the same visual object group. Concretely, for each visual object group $\mathcal{G}_S$ and each object $o \in S$, we normalize $\{r_{i,o}\}_{c_i \in \mathcal{G}_S}$ as follows:

$$A_{i,o}^{\text{int}} = \frac{r_{i,o} - \text{mean}_{j \in \mathcal{G}_S}[r_{j,o}]}{\text{std}_{j \in \mathcal{G}_S}[r_{j,o}] + \delta}, \quad c_i \in \mathcal{G}_S, \ o \in S,$$

where $\delta$ is a small constant for numerical stability. By construction, $A_{i,o}^{\text{int}}$ is computed within a naturally formed visual object group that refers to the same visual evidence. This within-group normalization establishes an explicit credit assignment mechanism for each piece of visual evidence. As a result, intrinsic advantages quantify the relative contribution of each visual object to the final answer, providing object-level signals for policy optimization and encouraging the policy to emphasize effective visual evidence.

### 4.4. Overall Policy Optimization

We integrate answer-level extrinsic advantages and object-level intrinsic advantages into a unified policy optimization step. Specifically, $A_i^{\text{ext}}$ is applied to all tokens in the reasoning chain, including both textual and coordinate tokens, while $A_{i,o}^{\text{int}}$ is only applied to the coordinate tokens corresponding to each referenced visual object $o \in S_i$, enabling object-level updates driven by within-group comparison.

We denote $\pi_{\theta_{\text{old}}}$ as the sampling policy and $\pi_\theta$ as the updated policy. Each candidate answer is written as a token sequence $c_i = (c_{i,1}, \ldots, c_{i,T_i})$ (including its reasoning chain). For each position $t$, the prefix context is $c_{i,1:t-1}$, consisting of the first $t-1$ tokens. The importance ratio is defined as:

$$\rho_{i,t}(\theta) = \frac{\pi_\theta(c_{i,t} \mid x, \ c_{i,1:t-1})}{\pi_{\theta_{\text{old}}}(c_{i,t} \mid x, \ c_{i,1:t-1})}.$$

We adopt the standard clipped surrogate objective with hyperparameter $\epsilon$ to promote stable policy updates:

$$g(\rho, A) = \min\left(\rho A, \ \text{clip}(\rho, 1-\epsilon, 1+\epsilon)A\right).$$

---

**Algorithm 1** Overall Policy Optimization with POLIA

---

**Require:** Initial policy $\pi_{\theta_{\text{old}}}$, task distribution $p(X)$, number of generations $N$, KL regularization coefficient $\beta$, clipping parameter $\epsilon$, extrinsic reward function weights $\{w_i\}$, intrinsic weight $\omega$
1: **for** each training iteration **do**
2:     Update old policy: $\theta_{\text{old}} \leftarrow \theta$;
3:     Sample input batch $x \sim p(X)$;
4:     Generate $N$ answers $\{c_1, \ldots, c_N\} \sim \pi_\theta(\cdot \mid x)$;
5:     Process grounding information and extract predicted bounding boxes;
6:     Compute weighted extrinsic rewards $\{R(c_i)\}$;
7:     Normalize rewards to compute extrinsic advantages $\{A_i^{\text{ext}}\}$;
8:     Group candidates $c_i$ referring to the same group of visual objects $\{S_i\}$ into visual object groups $\{\mathcal{G}_S\}$;
9:     Compute object-level intrinsic advantages $\{A_i^{\text{int}}\}$ using object confidence score $s_{i,o}$ within visual object groups $\{\mathcal{G}_S\}$;
10:    Combine advantages: $A_i \leftarrow A_i^{\text{ext}} + \omega A_i^{\text{int}}$;
11:    Update policy $\theta$ according to Equation 1;
12: **end for**

---

The overall training objective is:

$$\begin{aligned}
\mathcal{L}(\theta) = &-\mathbb{E}_x\Bigg[\frac{1}{N}\sum_{i=1}^{N}\sum_{t \in \mathcal{T}_i^{\text{all}}} g\Big(\rho_{i,t}(\theta), \ A_i^{\text{ext}} \\
&+ \omega \sum_{o \in S_i} \mathbb{I}\big(t \in \mathcal{T}_{i,o}^{\text{box}}\big) A_{i,o}^{\text{int}}\Big)\Bigg] + \beta \, \text{KL}\Big(\pi_{\theta_{\text{old}}} \parallel \pi_\theta\Big).
\end{aligned}$$
$$(1)$$

where $\mathcal{T}_i^{\text{all}}$ denotes the full reasoning-chain token set of $c_i$ and $\mathcal{T}_{i,o}^{\text{box}}$ denotes the coordinate tokens in $c_i$ that refer to object $o$, and $\omega$ is the intrinsic advantage weight. We present the pseudocode in Algorithm 1. See Appendix A.1 for more implementation details. In this way, extrinsic advantages encourage higher-reward answers under final-answer supervision, while intrinsic advantages provide explicit credit assignment signals on visual evidence through within visual object group comparison. This unified update preserves the efficiency of group-based RL and strengthens the learning signal for extracting and utilizing effective visual evidence.

## 5. Experiments

In this section, we evaluate POLIA on widely used multimodal reasoning benchmarks, focusing on the following research questions (RQs).

• **RQ1:** How does POLIA perform compared to representative baselines on multimodal reasoning benchmarks?

• **RQ2:** How do the extrinsic and intrinsic advantages influ-

*Table 1.* Performance comparison of POLIA with representative **Closed-Source** and **Open-Source** MLLMs on answer accuracy (ACC, %). The best scores are **bold** and the second best are underlined. *Average Improvements* denotes the average relative performance difference between POLIA and baseline models of the same parameter scale. $^+$ scores are taken from the respective models' official reports.

| Model | Physical Perception | | | Visual Mathematical Reasoning | | | Comprehensive Test |
|---|---|---|---|---|---|---|---|
| | *VSR* | *TallyQA* | *GQA* | *MathVista* | *MathVision* | *LogicVista* | *MME* |
| *Closed-Source Models* | | | | | | | |
| GPT-4o | 62.3 | 38.9 | 52.2 | 60.6 | 30.4$^+$ | 52.3 | 83.5 |
| Gemini2.5-pro | 64.0 | 49.8 | 60.5 | 54.1 | 73.3$^+$ | 73.8$^+$ | 92.9 |
| *Open-Source Models (7B)* | | | | | | | |
| Qwen2.5-VL | 41.3 | 48.0 | 33.9 | 48.8 | 25.1 | 44.5 | 92.9 |
| Qwen2.5-VL+DAPO | 60.7 | 52.8 | 58.8 | 61.9$^+$ | 27.3$^+$ | 47.5$^+$ | 92.2 |
| Qwen2.5-VL+GRPO | 59.0 | 48.0 | 58.2 | 65.5$^+$ | 26.3$^+$ | 47.1$^+$ | 92.6 |
| **POLIA** | **81.3** | **56.7** | **69.5** | **74.8** | **29.4** | **48.7** | **93.3** |
| *Average Improvements* | 27.6 ↑ | 7.1 ↑ | 19.2 ↑ | 16.1 ↑ | 3.2 ↑ | 2.3 ↑ | 0.7 ↑ |
| *Open-Source Models ($\leq$ 3B)* | | | | | | | |
| InternVL3-2B | 52.9$^+$ | 15.5$^+$ | 29.4$^+$ | 43.0$^+$ | 21.7$^+$ | 36.9$^+$ | 40.0$^+$ |
| Qwen2.5-VL-3B | 49.5 | 40.8 | 20.1 | 56.0 | 9.8 | 28.5 | 88.6 |
| Chain-of-Thought-3B | 37.5$^+$ | 33.2$^+$ | 39.5$^+$ | 33.0$^+$ | 20.0 | 38.1 | 41.3$^+$ |
| One-shot ICL-3B | 13.2$^+$ | 36.3$^+$ | 20.4$^+$ | 29.1$^+$ | 12.2 | 18.3 | 24.7$^+$ |
| Few-shot fine-tuning-3B | 59.7$^+$ | 44.5$^+$ | **64.6$^+$** | 45.0$^+$ | 12.8 | 17.0 | 68.3$^+$ |
| GRIT-3B | 61.2 | 43.6 | 57.9 | 56.2 | 12.3 | 39.4 | 85.4 |
| Qwen2.5-VL+DAPO-3B | 53.9 | 43.4 | 54.9 | 61.5 | 22.3 | 39.3 | 88.4 |
| Qwen2.5-VL+GRPO-3B | 53.5 | 41.6 | 57.1 | 62.4 | 24.4 | 38.5$^+$ | 88.3 |
| **POLIA-3B** | **71.9** | **48.8** | 60.2 | **63.9** | **24.5** | **40.4** | **89.2** |
| *Average Improvements* | 24.2 ↑ | 11.4 ↑ | 17.2 ↑ | 15.6 ↑ | 7.6 ↑ | 8.4 ↑ | 23.6 ↑ |

ence the learning dynamics of POLIA?

• **RQ3:** During training, does POLIA encourage visual object groups to become more stable and consistent?

• **RQ4:** How much additional computational cost does PO-LIA bring compared to GRPO?

### 5.1. Experiment Setup

**Datasets.** We evaluate POLIA on seven widely used multimodal reasoning benchmarks: VSR (Liu et al., 2023), TallyQA (Acharya et al., 2019), GQA (Hudson & Manning, 2019), MathVista (Lu et al., 2023), MathVision (Wang et al., 2024), LogicVista (Xiao et al., 2024), and MME (Fu et al., 2025). These benchmarks cover both physical perception (e.g., complex object counting and spatial relation understanding) and visual mathematical reasoning (e.g., mathematical and logical reasoning). More details on data preprocessing are provided in Appendix B.

**Baselines.** We compare POLIA with representative baselines that cover common training strategies for MLLMs. For the 7B setting, we adopt Qwen2.5-VL-7B-Instruct as the base model and include GRPO-style RL baselines (Qwen2.5-VL+GRPO, Qwen2.5-VL+DAPO) for fair comparison. For the 3B setting, we group baselines into two categories: (i) **Non-RL baselines**, including InternVL3-2B, Qwen2.5-VL-3B-Instruct, one-shot ICL, chain-of-thought,

and few-shot fine-tuning; (ii) **RL baselines**, including Qwen2.5-VL+GRPO-3B, Qwen2.5-VL+DAPO-3B, and GRIT-3B, which emphasizes visual evidence during optimization. We also report strong closed-source models (GPT-4o, Gemini2.5-pro) as reference points.

**Evaluation metrics.** Following prior work, we assess answer accuracy using GPT-based judges (Fan et al., 2025). Because model outputs are free-form natural language and may involve paraphrases, automated string matching is insufficient; instead, we adopt a GPT-as-a-judge protocol that focuses on semantic correctness rather than exact string match. Specifically, we use GPT-4o to compare the generated answer with the benchmark reference and produce an answer accuracy score in $[0, 1]$, where 0 indicates an incorrect answer and 1 indicates a fully correct answer. We report the average score on each benchmark as the main evaluation metric. Related details are provided in Appendix C.2.

### 5.2. Performance Comparisons with Baselines (RQ1)

Table 1 shows the comparison results. Closed-source models remain strong, while open-source base models lag behind, highlighting the difficulty of multimodal reasoning under outcome-only supervision. Notably, on benchmarks requiring precise visual grounding (VSR, GQA, TallyQA), POLIA-7B outperforms both GPT-4o and Gemini2.5-pro, reflecting its advantage in the effective use of visual evidence.

GRPO-style group-based RL (GRPO/DAPO) brings clear gains over the base model, yet their answer-level optimization provides limited credit assignment on visual evidence. In contrast, POLIA consistently surpasses GRPO/DAPO across benchmarks, especially on tasks requiring precise visual evidence. Concretely, for the 7B setting, POLIA improves VSR by +22.3%, TallyQA by +8.7%, and GQA by +11.3% over GRPO, while also boosting MathVista by +9.3%. For smaller models, POLIA-3B similarly outperforms RL baselines (e.g., VSR +18.4%; TallyQA +7.2%), whereas SFT is slightly higher on GQA, reflecting the advantage of direct answer supervision on that dataset. In addition, it is worth noting that POLIA is trained only on VSR and TallyQA with limited supervision, yet shows strong performance, as detailed in Appendix A.2. Nevertheless, the improvements observed across a broad range of evaluation tasks suggest that POLIA exhibits a certain degree of generalization beyond the training benchmarks.

Overall, these results indicate that strengthening visual-evidence credit assignment during policy optimization leads to robust gains on both perception-centric and reasoning-centric benchmarks, as reflected by MME.

### 5.3. Ablations on the Two Advantages (RQ2)

To examine how the extrinsic advantage and the intrinsic advantage influence the training dynamics of POLIA, we conduct ablation studies and track a normalized training reward that combines answer correctness and reasoning-format validity over training iterations on VSR and TallyQA.

As shown in Figure 3, consistent trends are observed on both datasets: removing the extrinsic advantage (w/o $A^{\text{ext}}$) leads to the lowest rewards and the slowest improvement, suggesting that extrinsic advantages are critical for stable optimization; removing the intrinsic advantage (w/o $A^{\text{int}}$) yields substantially better dynamics than removing the extrinsic advantage, but still trails the full method in both convergence speed and final reward plateau. This suggests that while the extrinsic advantage drives effective global optimization, the intrinsic advantage complements it by providing object-level guidance within each visual object group, improving credit assignment on visual evidence.

Generally, these results demonstrate that POLIA yields the most effective training dynamics by combining stable global learning with improved visual-evidence credit assignment.

### 5.4. Convergence Analysis on the Number of Visual Object Groups (RQ3)

We design this series of experiments to examine whether POLIA drives visual object groups to become more consistent during training, and whether such convergence co-evolves with accuracy. Here, sampled candidate answers

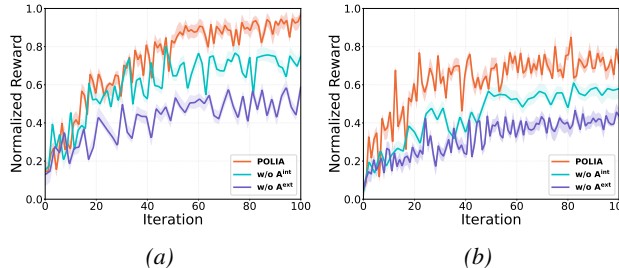

*(a)*          *(b)*

*Figure 3.* Training curves of the ablation study. The y-axis shows the normalized reward used for training. Curves are averaged over runs and reported with 95% confidence intervals. (a) Ablation results on VSR; (b) Ablation results on TallyQA.

are grouped by the visual object sets they rely on. The number of visual object groups is counted as the number of distinct visual object sets appearing among the sampled answers; a smaller value indicates that more answers rely on consistent visual evidence. We randomly sample from TallyQA with higher object counts, where visual evidence selection is more likely to be inconsistent at the beginning of training, making changes in visual object groups easier to observe. We track the evolution of the average number of visual object groups and answer accuracy over training iterations $\{0, 25, 50, 75, 100\}$ under different generation sizes $N \in \{8, 16, 24\}$. We further report the standard deviation of the number of visual object groups across samples to measure the stability of visual evidence selection.

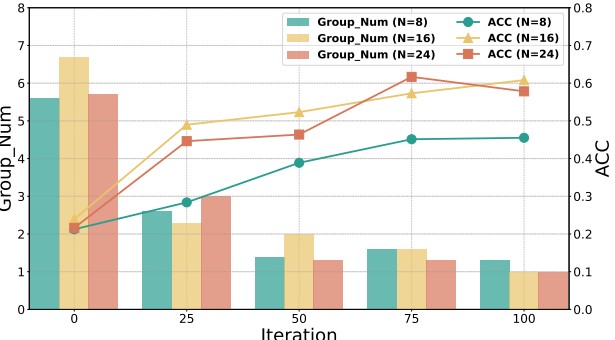

*Figure 4.* Evolution of the average number of visual object groups (left axis) and answer accuracy (right axis) over training iterations for different generation sizes ($N$).

**Relationships between visual object group numbers and accuracy.** As shown in Figure 4, at iteration 0, the number of visual object groups is high, indicating substantial disagreement among candidates on which objects to cite as visual evidence. As training proceeds, the number of visual object groups consistently decreases, dropping rapidly before iteration 50 and then stabilizing, suggesting that optimization compresses many plausible but inconsistent visual evidence into a smaller set of consistent ones. This behavior suggests that optimization progressively resolves ambigu-

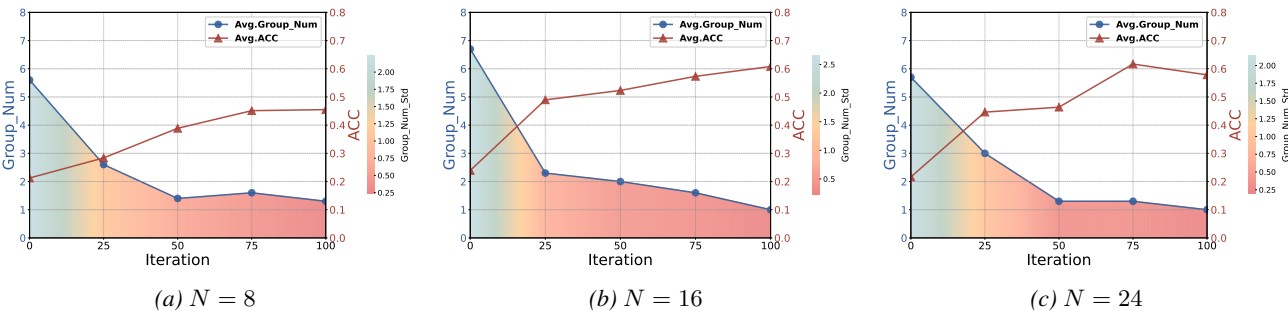

*Figure 5.* Stability of visual evidence selection. The shaded area indicates the standard deviation of the number of visual object groups across samples (left axis) over training iterations for different generation sizes ($N$), while accuracy is reported on the right axis.

ity in visual evidence selection, filtering multiple plausible but conflicting evidence choices into a smaller and more consistent set adopted by most candidates.

Meanwhile, ACC increases and saturates later, showing that visual object group convergence co-evolves with improved answer correctness. Varying the generation size further highlights an exploration–exploitation trade-off: $N = 8$ yields noticeably lower ACC than $N \in \{16, 24\}$, while $N \in \{16, 24\}$ are similar, indicating that insufficient exploration can limit performance but gains diminish once candidate diversity is adequate.

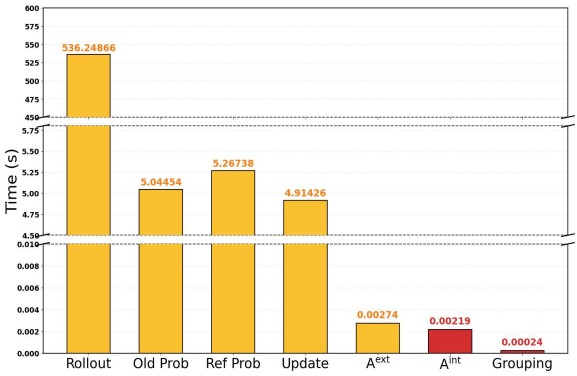

*Figure 6.* Per-iteration training time breakdown of POLIA. Yellow bars indicate components shared with the GRPO baseline, while red bars represent POLIA-specific additions ($A^{\text{int}}$ and Grouping). The y-axis uses two broken scales to accommodate small values.

**Stability of visual evidence selection.** As shown in Figure 5, across all $N$, the standard deviation of the number of visual object groups decreases together with its mean, indicating that evidence disagreement becomes less volatile across different samples and that evidence selection is more predictable and robust. The reduction in standard deviation is larger in the early stages of training and gradually tapers off later, which is consistent with a two-phase optimization process: in the early phase, the model rapidly suppresses highly inconsistent evidence that leads to divergent reasoning behaviors; in the later phase, as the number of visual

object groups stabilizes, training primarily focuses on resolving remaining ambiguous cases, with changes becoming smaller as convergence is approached. Finally, by ablating intrinsic advantage at $N = 8$, we observe that the number of visual object groups increases by 50% compared to the full method, suggesting that intrinsic advantage plays an important role in promoting visual object group convergence and stable evidence selection.

### 5.5. Computational Cost Analysis (RQ4)

To show the efficiency of POLIA, we decompose the training process into several parts. Our approach builds upon the GRPO framework, maintaining a critic-free architecture but introducing two additional steps: *Grouping* and the computation of $A^{\text{int}}$ (as detailed in Section 4.3). We benchmark the per-iteration training time of POLIA-3B. As illustrated in Figure 6, the Rollout phase is the dominant bottleneck, consuming 536.25 s (97.24%) of the total time. This high latency is primarily attributed to the *thinking* process, where the model must generate 2D bounding boxes. By contrast, the newly introduced components incur negligible overhead: the computation of $A^{\text{int}}$ accounts for only 0.002 s, while *Grouping* is virtually instantaneous. These results demonstrate that our method substantially enhances model performance with minimal impact on training efficiency.

### 6. Conclusions

The goal of multimodal reasoning is to enable models to reason correctly by leveraging visual evidence. However, existing group-based RL methods mainly optimize answer-level extrinsic rewards and lack explicit credit assignment over visual objects, which limits their ability to regulate how visual evidence is used during reasoning. In this work, we aim to address this limitation by improving credit assignment over visual objects in multimodal RL. We propose POLIA, a group-based RL method with intrinsic advantages defined on visual objects. Experiments on diverse multimodal reasoning benchmarks demonstrate its effectiveness. We believe much remains unexplored in multimodal RL.

## Acknowledgements

This work was sponsored by the Guangdong Basic and Applied Basic Research Foundation (No. 2025A1515010247), and the CCF-Kuaishou Large Model Explorer Fund (NO. CCF-KuaiShou 2025004), and the National Natural Science Foundation of China (No. 62506133).

## Impact Statement

This paper presents work whose goal is to advance the field of machine learning, particularly in multimodal reasoning and reinforcement learning. While advances in multimodal reasoning may have broader societal implications, such as improved performance in vision-language systems, we do not foresee any immediate ethical concerns or negative societal impacts specific to this work beyond those commonly associated with machine learning research.

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

# A. Experiment Details

## A.1. Experiment Setup

We use the AdamW optimizer with a learning-rate scheduler and set the learning rate to $2 \times 10^{-6}$. Experiments are run on NVIDIA A100 GPUs (40GB). For POLIA-3B, training uses a per-device batch size of 8 with a gradient accumulation step of 4. For each input, we sample a group of $N = 16$ candidate answers. For POLIA-7B, training uses a per-device batch size of 3 with a gradient accumulation step of 2. For each input, we sample a group of $N = 9$ candidate answers. To compute the intrinsic advantage, a predicted bounding box is considered successful if its Intersection over Union (IoU) exceeds 0.5. The reward signal is defined as a weighted sum, with weights of 0.7 for the IoU term and 0.3 for the L1 term. For the final advantage estimation, the coefficients for the sub-graph set advantage and GPT-based scoring are set to 1.0 and 1.5, respectively, while all other reward components use a default weight of 1.0.

## A.2. Data Usage

We provide additional clarification on the supervision setting used to train POLIA. In our experiments, POLIA is trained under a constrained object-level supervision setup, rather than relying on large-scale annotated data. Despite this constrained training setting, POLIA demonstrates consistent performance improvements across a diverse set of multimodal reasoning benchmarks. While we do not aim to make strong claims about data efficiency, this observation suggests that POLIA's learning behavior does not critically depend on extensive object-level annotations.

# B. Dataset Details

## B.1. VSR

VSR (Liu et al., 2023) contains tasks on spatial relation verification. Our training and evaluation data are obtained from the VSR subset of the Visual CoT benchmark (Shao et al., 2024a). The VSR subset, situated within the Relation Reasoning category, is of particular relevance to our work. It focuses on the fundamental task of spatial relation verification, which requires models to assess the validity of a spatial relationship (e.g., "left of," "inside," "under") between objects in a scene. For each VSR triplet (question-image-answer), the dataset provides precise ground-truth bounding boxes for the entities involved. This ensures that a model must demonstrate accurate visual grounding—identifying the exact location of objects—before performing high-level spatial reasoning. The task challenges models to resolve complex and often subtle spatial configurations in diverse real-world images.

To improve training quality, we performed several manual adjustments to the dataset. We first filtered out samples containing ambiguous or subjective answers to ensure a deterministic training signal. Additionally, we slightly adjusted the dimensions of the bounding boxes within the training set. This modification assists the model in better capturing target objects, thereby enhancing its visual grounding performance during the training process. During the evaluation phase, the model is provided only with question-image-answer triplets. No bounding box information is available during inference. This setup forces the model to rely entirely on its learned grounding and reasoning capabilities to verify spatial relations without any external visual cues. As shown in Table 1, we utilize a small amount of data sourced from a certain paper (Fan et al., 2025).

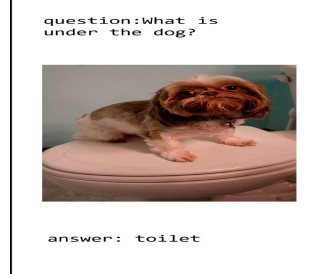 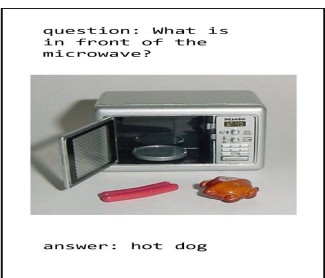 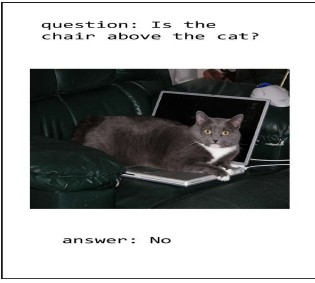

*Figure 7.* Illustrative examples from the VSR dataset.

## B.2. TallyQA

TallyQA (Acharya et al., 2019) is a benchmark designed to evaluate open-ended counting capabilities within the context of Visual Question Answering (VQA). Unlike standard counting tasks that primarily rely on basic object detection, TallyQA distinguishes between simple and complex counting questions. The complex category, which is of particular interest to our study, requires models to reason about intricate relationships between objects, identify specific visual attributes, and integrate contextual information (e.g., How many striped containers are to the left of the sink?). By utilizing diverse images from Visual Genome and VQA v2.0, the benchmark provides a rigorous environment for assessing a model's ability to perform high-level numerical reasoning and fine-grained visual grounding in real-world scenes.

Consistent with our approach for VSR, we apply the same processing methodology to the TallyQA dataset to enhance the model's grounding and reasoning capabilities while reducing label noise. For the training set, we manually filter out ambiguous samples and resize bounding boxes to better supervise the model's attention. During the evaluation phase, we utilize only the question-image-answer triplets, requiring the model to verify numerical relations without the aid of bounding box information. As shown in Table 1, we utilize a small amount of data sourced from a certain paper (Fan et al., 2025).

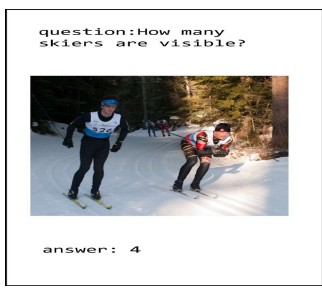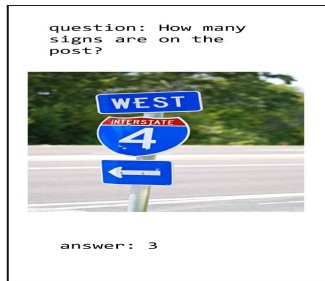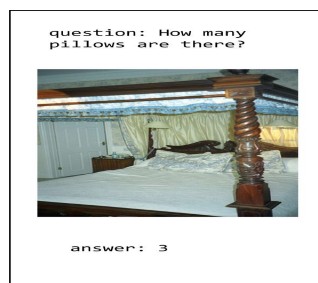

*Figure 8.* Illustrative examples from the TallyQA dataset.

## B.3. GQA

GQA (Hudson & Manning, 2019) is a large-scale benchmark designed for real-world visual reasoning and compositional question answering. Unlike traditional VQA datasets, GQA leverages the structured Scene Graphs of Visual Genome to create questions with complex logical dependencies. These questions require models to perform multi-step reasoning, including object localization, attribute identification, and the understanding of intricate spatial relationships. The dataset's strength lies in its ability to evaluate whether a model can truly reason through a scene rather than relying on statistical biases.

Following a simple manual refinement, we filter out ambiguous samples to maintain data quality. During evaluation, the model is provided solely with the image and question. As shown in Table 1, we utilize a small amount of data sourced from a certain paper (Fan et al., 2025).

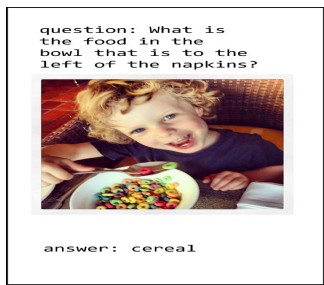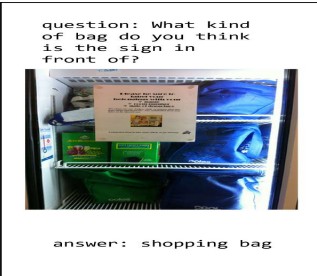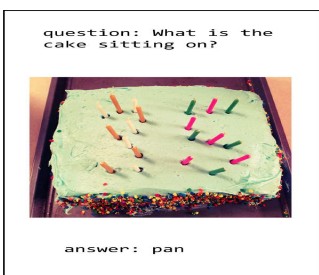

*Figure 9.* Illustrative examples from the GQA dataset.

### B.4. MathVista

MathVista (Lu et al., 2023) is a comprehensive benchmark specifically designed to evaluate mathematical reasoning within visual contexts. It integrates 6,141 examples by consolidating 28 existing multimodal datasets and introducing three new specialized ones: IQTest, FunctionQA, and PaperQA. The benchmark covers a wide array of mathematical domains, including basic arithmetic, geometry, algebraic reasoning, and statistical analysis, all presented through diverse visual forms such as charts, diagrams, and geometric shapes. Unlike standard VQA tasks, MathVista requires models to perform fine-grained visual perception combined with multi-step compositional reasoning, posing a significant challenge to current MLLMs.

We utilize MathVista solely for testing purposes without any training involvement. By retaining only the essential visual and textual components (images, questions, and answers), we challenge the model to tackle complex mathematical tasks independently. The evaluation is conducted without external prompts or spatial hints to ensure the results reflect the model's genuine reasoning depth. As shown in Table 1, we utilize a small amount of data sourced from certain papers (Fan et al., 2025; Wang et al., 2025d).

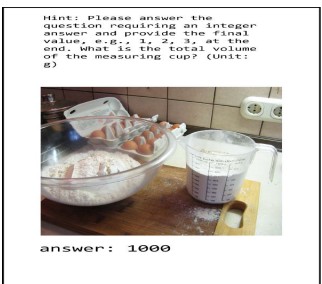 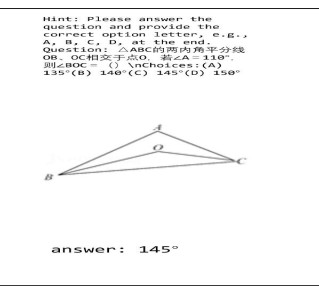 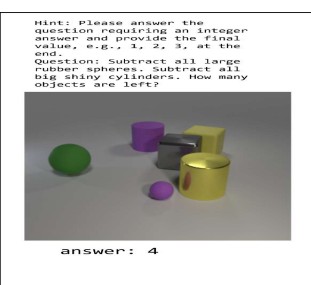

*Figure 10.* Illustrative examples from the MathVista dataset.

### B.5. MathVision

MathVision (Wang et al., 2024) is a meticulously curated visual mathematical benchmark consisting of 3,040 high-quality problems collected from real-world mathematical competitions. The dataset spans 16 distinct logical disciplines—including plane and solid geometry, function transformation, and combinatorics—categorized into five difficulty levels. Unlike previous benchmarks, MathVision emphasizes the integration of complex mathematical theory with diverse visual representations. Each problem requires models to accurately perceive intricate geometric structures or mathematical notations and execute multi-step logical reasoning to reach a solution, providing a rigorous test for the mathematical reasoning capabilities of modern MLLMs.

For MathVision, our experiments are confined to the evaluation phase. The dataset is reduced to its most basic form—triplets of images, queries, and ground truth to eliminate any potential information leakage from metadata. Consequently, the model must navigate through the reasoning process autonomously, with no access to bounding boxes or supplementary hints during inference. As shown in Table 1, we utilize a small number of data sourced from certain papers (Wang et al., 2024; 2025d; Zhu et al., 2025).

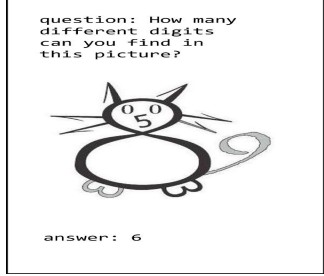 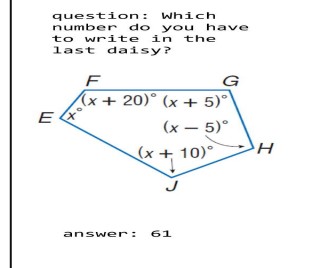 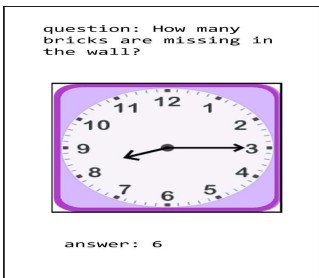

*Figure 11.* Illustrative examples from the MathVision dataset.

## B.6. LogicVista

LogicVista (Xiao et al., 2024) is an evaluation benchmark focused on the logical reasoning capabilities of Multimodal Large Language Models in visual contexts. It consists of 448 high-quality multiple-choice questions covering 5 core logical reasoning tasks and 9 distinct capabilities, such as deductive reasoning, numerical logic, and spatial puzzles. The dataset is uniquely designed to evaluate how models integrate visual perception with complex cognitive tasks, like navigation and puzzle-solving. Each entry is meticulously annotated with human-written reasoning chains, providing a rigorous standard for assessing both the accuracy and the logical consistency of MLLMs in challenging real-world scenarios.

LogicVista is employed exclusively as a blind test set. We simplify the input to basic image-question pairs to evaluate the model's logical consistency. To maintain a fair and challenging environment, the model is prohibited from using any bounding box assistance, requiring it to interpret the visual context and logical constraints entirely on its own. As shown in Table 1, we utilize a small amount of data sourced from certain papers (Wang et al., 2025d; Zhu et al., 2025).

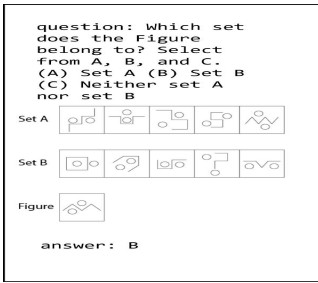 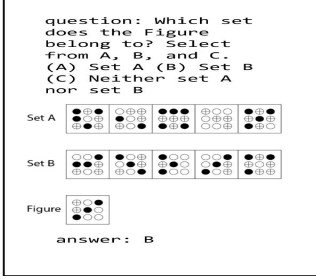 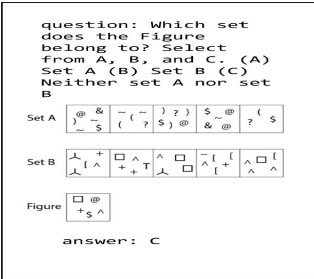

*Figure 12.* Illustrative examples from the LogicVista dataset.

## B.7. MME

MME (Fu et al., 2025) is a comprehensive benchmark designed to evaluate the multifaceted capabilities of Multimodal Large Language Models (MLLMs). It covers a total of 14 subtasks, systematically categorized into two dimensions: perception and cognition. The perception dimension evaluates the model's ability to recognize objects, counting, and spatial positions, while the cognition dimension challenges the model with higher-level tasks such as commonsense reasoning, numerical calculation, and code recognition. A distinctive feature of MME is that all instruction-answer pairs are manually designed rather than sourced directly from public datasets, which effectively prevents data leakage and ensures a more rigorous assessment of a model's true generalization ability.

In our study, MME is used exclusively for evaluation. We manually filter the dataset to ensure all test samples provide clear and deterministic signals. During the evaluation phase, the model is provided with only the image and the question. It must rely entirely on its own perception and reasoning capabilities to generate the answer, without the assistance of bounding boxes or any additional visual cues. As shown in Table 1, we utilize a small amount of data sourced from certain papers (Fan et al., 2025).

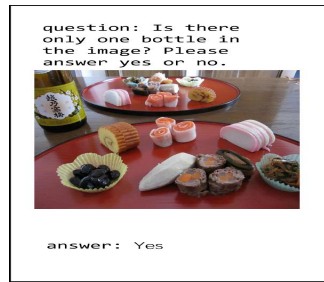 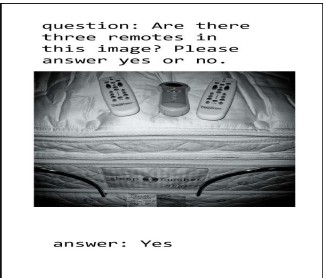 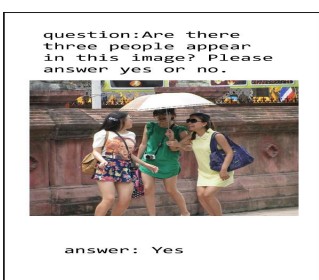

*Figure 13.* Illustrative examples from the MME dataset.

## C. Prompts

### C.1. Prompts for Generating Bounding Box

As illustrated in Figure 14, we design a specific prompt to facilitate the generation of bounding boxes, thereby effectively integrating the model's grounding and reasoning capabilities. Specifically, we include the explicit instruction "2D bounding boxes" to guide the model in generating spatial coordinates. Furthermore, by providing the format template [x_min, y_min, x_max, y_max], we ensure the model adheres to the correct output rules and more accurately captures objects within the image. This prompting strategy is consistently applied across both the training and testing phases.

---

### Prompt for Generating Bounding Boxes

{question}

**Guidelines:**

First, think between <think> and </think> while providing all potentially useful **2D bounding boxes** in JSON format under the key **'bbox_2d'** as **[x_min,y_min,x_max,y_max]**.
Then, based on the thinking contents and coordinates, rethink between <rethink> </rethink> and then answer the question using only one word or a single numeric value, placing your answer after <answer>.

---

*Figure 14.* The template instructs the model to first generate 2D bounding boxes in a specific JSON format [x_min, y_min, x_max, y_max] within the ⟨ think⟩ tags. Subsequently, the model is guided to rethink its logic based on these spatial coordinates before providing a concise final ⟨answer⟩ after the answer tag.

### C.2. Prompt for Scoring

Evaluation via GPT-4o is conducted as follows. To ensure a robust and objective assessment, we utilize GPT-4o as an automated evaluator to score the model's performance. Specifically, we adopt the evaluation prompt from GRIT (Fan et al., 2025), which facilitates a comparison between the model's output and the ground-truth answer. The detailed scoring template and criteria are illustrated in Figure 15. This approach allows for a consistent and fine-grained analysis of the model's reasoning accuracy.

## D. More Examples

## Prompt for Scoring

**Guidelines:**

You are responsible for proofreading the answers, you need to give a score to the model's answer by referring to the standard answer, based on the given question. The full score is 1 point and the minimum score is 0 points. Please output the score in the json form "{{score: <score>}}". The evaluation criteria require that the closer the model's answer is to the standard answer, the higher the score.

Question: {question}

Standard answer: {answer}

Model's answer: {predicted_content}

*Figure 15.* The scoring prompt for GPT-4o-based evaluation. Following the protocol in GRIT (Fan et al., 2025), this prompt instructs GPT-4o to act as an impartial judge. It evaluates the alignment between the model's prediction and the ground truth, assigning a score from 0.0 to 1.0 based on the correctness and completeness of the reasoning.

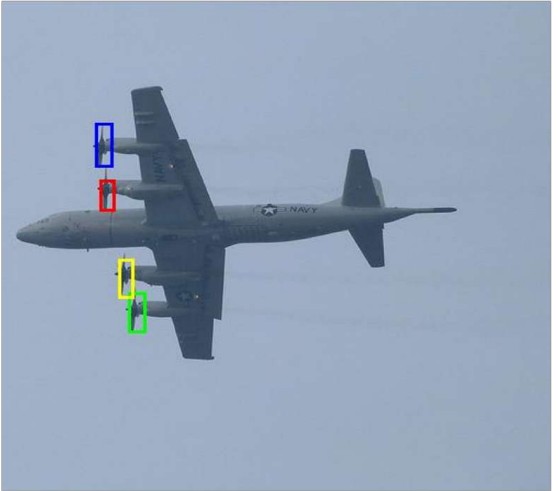

Question: How many propeller are there?
Ground Truth Answer:4

**Model Output**

```
<think>
{"bbox_2d": [85, 124, 101, 169]
, "label": "propeller"},
{"bbox_2d": [89, 182, 103, 214]
,"label": "propeller"}
{"bbox_2d": [115, 297, 131, 338]
,"label": "propeller"},
{"bbox_2d": [106, 263, 120, 303]
,"label": "propeller"}
</think>
```

```
<rethink>
The airplane has four propellers.
</rethink>
<answer>4</answer>
```

*Figure 16.* During the Think phase, the model generates object detection bounding boxes to identify relevant components. In the Rethink phase, it synthesizes this spatial information to form a logical conclusion, ultimately producing the final answer.

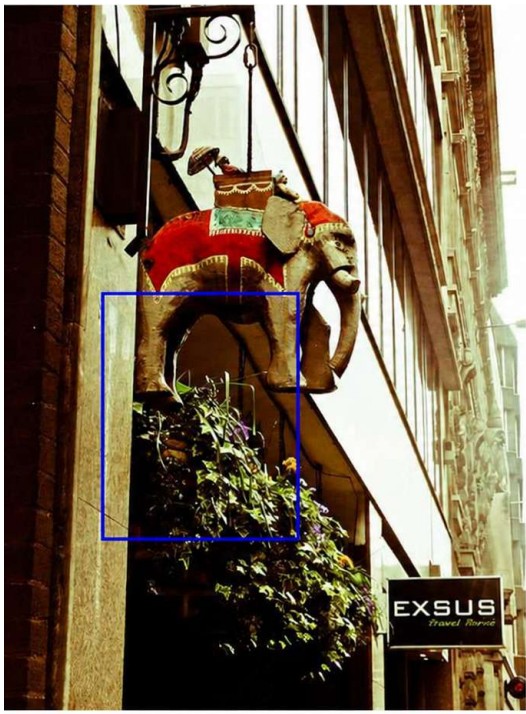

Question: Is the potted plant below the elephant?
Ground Truth Answer: Yes

**Model Output**

```
<think >
{"bbox_2d": [89, 260, 271, 485],
"label": "potted plant"}
</think>
```

```
<rethink>
The potted plant is directly below the
elephant.
</rethink>
<answer>yes</answer>
```

*Figure 17.* Case study on visual spatial reasoning. This example illustrates how the model utilizes bounding boxes to ground objects before performing logical deduction to verify spatial relationships, showcasing the integration of localization and reasoning.

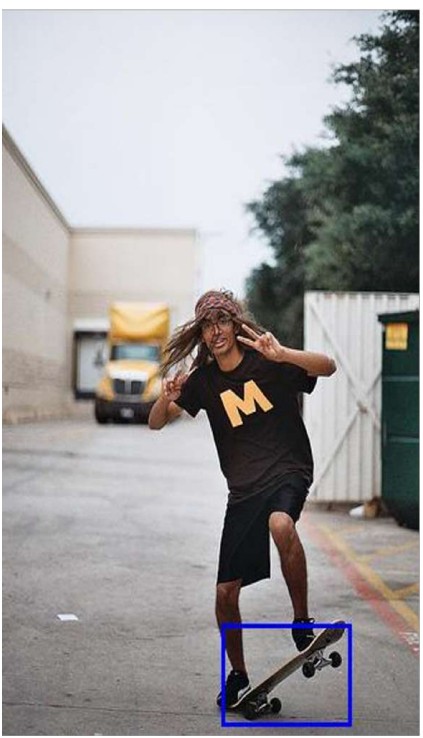

Question:What is the man on?
Ground Truth Answer:skateboard

**Model Output**

```
<think>
The relevant bounding box information is:
bbox_2d: [175, 423, 278, 493]
<think>
```

```
<rethink>
man is performing a trick on his
skateboard, which is why he's airborne.
The skateboard is positioned underneath
him, with its wheels facing upwards due to
the motion.
</rethink>
<answer>skateboard</answer>
```

*Figure 18.* The figure illustrates how the model utilizes 2D bounding boxes to focus on key entities and leverages its reasoning tags (⟨think⟩, ⟨rethink⟩) to resolve the relationship between the man and the skateboard in a dynamic context.

