# OpenReview forum: "POLIA: Policy Optimization with Visual-Object-Level Intrinsic Advantage for Multimodal Reasoning"
_ICML.cc/2026/Conference — ICML 2026 regular_

### Official Review · Reviewer_Yy3B · 2026-03-07

**Soundness:** 4
**Presentation:** 4
**Significance:** 3
**Originality:** 4
**Overall Recommendation:** 4
**Confidence:** 4

**Summary:**

This paper introduces POLIA, a reinforcement learning method for multimodal reasoning that improves how models use visual evidence. Standard group-based RL methods reward only final answers, so models may ignore key objects or rely on wrong visual cues. POLIA adds a visual-object-level intrinsic advantage to the usual answer-level extrinsic advantage. It groups candidate responses by the objects they reference, scores object confidence from bounding-box quality, and assigns finer-grained credit to relevant visual evidence. Across seven benchmarks, POLIA outperforms GRPO/DAPO baselines, especially on grounding-heavy tasks, while adding almost no extra computational cost during training and improves evidence consistency over time significantly.

**Compliance With Llm Reviewing Policy:**

Affirmed.

**Key Questions For Authors:**

See above.

**Limitations:**

yes

**Strengths And Weaknesses:**

# Strengths
1. The paper identifies a real limitation of existing group-based RL methods for multimodal reasoning: they mainly optimize at the answer level and lack explicit credit assignment over visual evidence. POLIA frames this as an object-level credit assignment problem, which is highly relevant to grounding failures and hallucination in MLLMs.
2. POLIA combines answer-level extrinsic advantage with visual-object-level intrinsic advantage, and applies the object-level signal specifically to coordinate tokens corresponding to referenced objects. This makes the optimization more fine-grained than standard outcome-level RL and provides a clearer mechanism for how the model learns to rely on relevant visual evidence.
3. Beyond the main benchmark results, the paper includes ablations on the two advantage terms, an analysis of visual object group convergence, and a computational cost breakdown. The results suggest that POLIA improves grounding-heavy multimodal reasoning tasks while adding very little overhead beyond standard GRPO training.

# Weaknesses
1. POLIA relies on predicted bounding boxes, grouping by object sets, and computing intrinsic advantages over those referenced objects. This is a natural fit for tasks such as spatial reasoning and counting, but it is less clear whether the same formulation extends well to multimodal reasoning settings where evidence is not naturally object-box based, such as charts, abstract diagrams, or more diffuse visual semantics.
2. The appendix states that the method is trained under a constrained object-level supervision setup, and that the authors manually filtered ambiguous samples and adjusted bounding boxes in the training data. While this does not invalidate the method, it does make it harder to disentangle how much of the gain comes from the algorithm itself versus the data curation and supervision assumptions.
3. Although the paper shows transfer from training on VSR and TallyQA to a broader set of evaluation benchmarks, the strongest gains appear on tasks that are closely tied to precise visual grounding. As a result, it is still unclear how robustly POLIA generalizes to more open-ended, abstract, or non-object-centric multimodal reasoning problems.
4. The text in the dataset visualizations in the appendix is overly compressed, which makes it difficult to read.

---

> ### Author Rebuttal · Authors · 2026-03-30
>
> Hi Reviewer Yy3B,
>
> Thank you for your valuable time and detailed review. We sincerely appreciate your overall positive assessment of our work, including your recognition of the design of POLIA for multimodal reasoning, the extensive evaluation across multiple benchmarks, as well as its effectiveness with minimal additional computational cost. Your questions are very helpful in further improving the paper, particularly regarding the extensibility of the method and its generalization ability. We have conducted additional analysis and clarifications in response to your comments. Below we address each point in detail.
>
>
> ## Q1. Can POLIA Extend to Abstract Settings
>
> >**TLDR:** We agree that multimodal tasks differ in visual evidence structure and POLIA **can be extended to abstract settings**. Moreover, our evaluation already includes abstract visual math benchmarks, where **POLIA remains effective**.
>
> >**Action:** Your comments are valuable for strengthening our paper, we will add analysis of different task types in Section 5.2 to improve completeness and clarity.
>
> POLIA can be extended to more general visual representations beyond bounding boxes. Specifically, bounding boxes can be viewed as one instantiation of **parseable visual units**; by extending the notion of units to other structured visual elements (e.g., regions or patches), the intrinsic advantage mechanism in POLIA can be directly applied by redefining the unit representation accordingly. The experiments on Visual Mathematical Reasoning datasets demonstrate POLIA’s extensibility to abstract settings, where visual evidence is abstract and dispersed. As shown in Table 1, POLIA-3B achieves consistent average improvements on MathVista, MathVision, and LogicVista (15.6, 7.6, and 8.4, respectively), demonstrating the effectiveness and broader applicability of our approach.
> ## Q2. Whether Data Preprocessing Affords Extra Gains
>
> >**TLDR:** All methods (incl. baselines) use the same data and pipeline. Only a small portion of data is slightly corrected to fix clear label noise, ensuring fair and valid evaluation—**not extra gains**.
>
> We agree that data design can affect results, so we ensure strict fairness. All methods share identical train/test sets and the same preprocessing. Thus, performance gaps come from the methods, not data handling. Ambiguous samples noted in the Appendix are mainly corrected to address obvious label bias, so training signals better reflect true prediction quality, ensuring valid training and evaluation.
> ## Q3. Can POLIA Generalize to Non-Object-Centric Settings
>
> >**TLDR:** Beyond grounding tasks, we evaluate on abstract multimodal benchmarks such as MathVista, MathVision, and LogicVista. Results show steady gains over baselines across tasks and abstraction levels, indicating **good cross-task generalization**.
>
> >**Action:** We appreciate your insightful comments, we will further clarify task types and results in Section 5.2 to better present the method’s generalization ability.
>
> POLIA can generalize to non-object-centric scenarios, where it treats elements such as formulas, chart components as valid visual evidence units, enabling precise credit assignment to these key semantic components. In addition to Physical Perception datasets like VSR and TallyQA, we also evaluate on more abstract multimodal benchmarks, including MathVista, MathVision, and LogicVista. These datasets involve chart understanding, mathematical reasoning, and logical inference. As shown in Table 1, POLIA achieves consistent improvements over baselines across these tasks. This indicates that the gains are not limited to object-centric tasks but also extend to broader multimodal reasoning settings.
> ## Q4. Appendix Figure Readability
>
> >**Action:** Thank you for the feedback and we will redesign and optimize the related figures in the revision to improve readability and information presentation.
>
> We agree that some appendix figures are information-dense with crowded text, which affects readability. We sincerely thank you for the detailed suggestion, which helps improve the overall clarity of the paper.
> ##
> >We look forward to further discussion and remain available to address any additional questions. Thank you again for helping us improve the paper, and we hope our clarifications alleviate your concerns. We would be grateful if this could be reflected in your updated scores.

---

> > ### Author Rebuttal · Reviewer_Yy3B · 2026-04-03
> >
> > Thanks for the detailed reply from the author. My concerns have been addressed.

---

> > > ### Author Response · Authors · 2026-04-06
> > >
> > > We sincerely thank you for your careful review and for confirming that our concerns have been adequately addressed. We appreciate the time and effort you devoted to evaluating our responses.
> > >
> > > We are encouraged by your positive evaluation, and we appreciate your constructive comments. Your comments have truly informed and guided our thinking on follow-up research. We value the careful consideration reflected in your review. Given that the concerns have been fully resolved, we would be grateful if you would consider updating your score accordingly. Thank you again for your time and consideration.

---

### Official Review · Reviewer_NHMm · 2026-03-08

**Soundness:** 3
**Presentation:** 3
**Significance:** 3
**Originality:** 3
**Overall Recommendation:** 4
**Confidence:** 2

**Summary:**

This paper presents POLIA, a group-based reinforcement learning method designed to improve multimodal reasoning in large language models by introducing a two-stage advantage computation mechanism. The core observation is that existing RL methods like GRPO compute advantages purely at the answer level, which fails to provide meaningful credit assignment when different candidate answers rely on different visual evidence. POLIA addresses this by first computing standard answer-level extrinsic advantages, and then constructing visual object groups within which object-level intrinsic advantages are computed using IoU-based confidence scores. Experiments on seven multimodal benchmarks demonstrate consistent improvements over GRPO-style baselines.

**Compliance With Llm Reviewing Policy:**

Affirmed.

**Final Justification:**

Authors' response and clarification improve this work. I believe their revised version will meaningfully improve the quality of this work. Therefore, I have increased my score.

**Key Questions For Authors:**

* The intrinsic advantage computation relies on matching predicted bounding boxes to ground-truth object annotations. How sensitive is POLIA to the quality and coverage of ground-truth object annotations? Would the method degrade significantly on datasets with noisy or incomplete annotations, and how does the approach generalize to settings where such annotations are unavailable?
* The visual object grouping uses exact set equality ($S_i = S_j$) over matched object identities. In practice, how often do singleton groups arise, e.g., only one candidate refers to a given object set, and how does POLIA handle such cases given that within-group normalization is undefined or degenerate for groups of size 1?
* The experiments are conducted exclusively on Qwen2.5-VL (3B and 7B) and a few frozen baselines using InternVL3-2B. Given the rapid development of MLLM architectures, it is unclear whether POLIA's gains hold for more recent model families such as Qwen3-VL or InternVL3.5. Can the authors provide some experimental evidence on whether the method's benefits are architecture-agnostic, or whether they depend on specific properties of the base models tested?
* The object confidence score is computed as a weighted combination of IoU (weight 0.7) and normalized L1 distance (weight 0.3), clipped to [0,1]. Were these weights validated through ablation? How does performance change with alternative formulations (e.g., IoU-only, or a learned confidence score), and how sensitive are the results to this hyperparameter?
* The overall objective includes a scalar weight ω balancing extrinsic and intrinsic advantages. How was ω selected, and how sensitive is performance to its value? A sensitivity analysis would help clarify whether careful tuning of ω is required to apply POLIA to new settings.

**Limitations:**

The paper includes a brief impact statement and partially acknowledges limitations through discussion of data usage. However, several important limitations are not adequately discussed, such as the requirement for ground-truth bounding box annotations during training, the potential for degenerate grouping behavior (e.g., singleton groups), the narrow scope of the evaluation base model (primarily Qwen2.5-VL), etc. It may be more helpful if the authors can add a limitations section to discuss and address these points explicitly.

**Strengths And Weaknesses:**

**Strengths**
* The problem framing is technically well-motivated. The observation that answer-level rewards cannot fairly compare candidates relying on different visual evidence is sound and clearly articulated. The two-stage advantage design logically follows from this problem.
* The paper is clearly presented, with Figure 1 showing a high-level contrast with GRPO, and Figure 2 giving a detailed walkthrough of the full pipeline.
* The combination of group-level extrinsic advantages with within-group intrinsic advantages computed over visual objects is a well-reasoned extension of GRPO.

**Weaknesses**
* The confidence score computation (IoU + L1 weighting, clipped to [0,1]) is presented without ablation. It is unclear whether the specific weighting (0.7/0.3) is principled or tuned, and whether alternative formulations would significantly change results.
* The grouping mechanism depends on exact set equality of matched object identities (Si = Sj), which could be brittle in practice. For example, small differences in predicted bounding boxes leading to different matched ground-truth objects could fragment groups unnecessarily or yield singleton groups that provide no within-group normalization signal.
* The theoretical properties of the combined objective (Equation 1) are not analyzed. In particular, it is not clear whether the combined advantage signal remains unbiased or introduces any systematic gradient direction that could interfere with learning.
* The notation in Section 4.3 introduces intrinsic advantage $A^{int}_{i,o}$ per object $o$, but in the optimization objective (Equation 1) it appears as a sum over objects $o \in S_i$ without explicit clarification of how per-object advantages are aggregated into a per-token signal.
* The appendix provides useful dataset details, while does not include the training data statistics, such as the number of samples used per dataset, making it difficult to assess data efficiency claims.
* POLIA requires ground-truth bounding box annotations at training time, which restricts applicability to datasets where such spatial annotations exist.
* The POLIA seems to be implemented exclusively on Qwen2.5-VL at the 3B and 7B scales, which are relatively older model versions. It is unclear whether POLIA's advantages smoothly transfer to more recent and capable architectures such as Qwen3-VL or InternVL3.5, which may have substantially different internal visual grounding behaviors. Without experiments or discussion on newer model families, it is difficult to assess whether the observed gains reflect a general property of the method or are specific to the architectural characteristics of the tested models.

---

> ### Author Rebuttal · Authors · 2026-03-30
>
> Hi Reviewer NHMm,
>
> Thank you for your valuable time, your feedback helps improve the depth of our analysis. We provide our replies below.
> ## Q1. Whether POLIA Is Sensitive to Annotation Quality and Coverage
> >**TLDR:** We agree annotation quality will certainly affect performance, thus we **improve robustness to annotations** via matching-based filtering and confidence modeling.
>
> >**Action:** We will add discussion on imperfect annotations in Section 4.3.
>
> **Robustness to annotation noise:** POLIA does not use all annotations; it only retains predictions with reliable spatial alignment via IoU matching and threshold-based filtering, thus filtering out low-quality annotations. The confidence score further accounts for overlap and location shift, reducing the impact of unstable matches on advantage estimation. Thus, training stays stable under moderate noise without over-reliance on inaccurate annotations.
>
> **Impact of incomplete annotation coverage:** When annotations are incomplete, some predicted boxes may not match any ground truth and are filtered as “nomatch”. This acts as a conservative selection of visual evidence, using only semantically aligned objects. Although it reduces available supervision, it avoids adding wrong or uncertain regions, thus reducing the impact of incomplete annotations.
>
> **Generalization to unlabeled settings:** POLIA’s core—advantage assignment over interpretable visual units—does not rely on specific annotation formats. Without explicit annotations, it can be extended to use model-generated visual units (e.g., regions or patches), supporting weakly supervised or unsupervised settings.
> ## Q2. How Often Do Singleton Groups Occur and Is Special Handling Necessary
> >**TLDR:** Singleton groups are rare in practice and add no extra advantage scaling; thus, their overall impact is **limited**.
>
> Statistics show singleton groups occur less than 6.3% in our experiments, as multiple candidate answers often share visual evidence. In these rare cases, normalization adds no extra advantage scaling, keeping numerical stability. They also appear only once without repeated support, thus are less reliable. As POLIA optimization relies on the statistical behavior of group structures, a few singleton groups have little impact on training dynamics. Instead, this treatment helps avoid unstable signals from small groups and raises training stability.
> ## Q3. Latest Architecture Extension
> >**TLDR:** We report results on Qwen3-VL-8B/4B with same metrics as Table1, showing **POLIA outperforms the baselines**.
>
> >**Action:** Thank you for your advice, we will include the results in Section 5.2.
>
> Hardware: NVIDIA A800-80GB (NVLink); 8B uses 3 cards, 4B uses 2 cards. Hyperparameters: All settings follow Sec. A.1, no extra tuning. Dataset & evaluation: VSR and TallyQA, same splits and GPT‑4o judge scoring as Table 1.
> |Model|VSR|TallyQA|
> |-|-|-|
> |Qwen3-VL-8B|68.5|55.9|
> |POLIA-8B|**75.2**|**77.0**|
> |Qwen3-VL-4B|65.9|54.4|
> |POLIA-4B|**74.7**|**61.6**|
> ## Q4&Q5. Sensitivity Analysis
> We sincerely thank you for this valid question. Additional experiments show all hyperparameters vary smoothly over wide ranges, showing low sensitivity, with chosen values in stable, near-optimal regions. The intuitive curve plots:https://postimage.me/image/pdzi, https://postimage.me/image/pdhY. This concern is also addressed in Q4.Sensitivity Analysis of our reply to Reviewer jG4W; we respectfully refer you there for details, as this allows us to devote more space to addressing your other valuable concerns.
> ## Other Minor Concerns
> >**TLDR:** For concerns that you did **not mark as critical** but raised in weaknesses, we would like to provide clarifications as follows. We sincerely appreciate your feedback and will improve the presentation and theoretical exposition in the revision.
>
> Grouping Mechanism Sensitivity: In grouping, Si denotes object sets from IoU matching and thresholding, not exact box equality of raw bounding boxes. This allows spatial tolerance, so small box variations do not split objects, improving robustness to detection noise.
>
> Objective Function Bias Analysis: The overall advantage is a weighted combination of A_ext and A_int. A_ext gives an unbiased signal based on overall quality, while A_int complements and refines it at object level. From an optimization view, this is linear weighting integrating multi-granularity information in a unified framework, not a new objective. Experiments also show steady gains across benchmarks without instability from signal conflicts.
>
> Advantage Explanation: We would like to note that the sum over objects o∈Si only enumerates all objects, while the indicator acts as a selector. For each token t, only the term for its associated object is active; thus, each object’s intrinsic advantage applies only to its corresponding tokens.
>
> >Thank you again for helping us improve the paper. We sincerely hope that our clarifications address your concerns and may positively affect the score.

---

> > ### Author Rebuttal · Reviewer_NHMm · 2026-04-01
> >
> > Thank you for your detailed response, and I believe your revision will further strengthen the quality of this work. I will increase my score accordingly.

---

> > > ### Author Response · Authors · 2026-04-06
> > >
> > > We sincerely thank you for the time and effort you devoted to evaluating our work, and for updating your assessment. We greatly appreciate the constructive nature of your comments. We are very encouraged by your positive evaluation, and we are grateful that our clarifications and revisions have helped address your concerns.
> > >
> > > We truly appreciate your support and insightful suggestions, and we are thankful for the opportunity to further improve our work based on your comments.

---

### Official Review · Reviewer_Jtum · 2026-03-12

**Soundness:** 3
**Presentation:** 2
**Significance:** 3
**Originality:** 3
**Overall Recommendation:** 4
**Confidence:** 4

**Summary:**

This paper introduces a visual-object-level policy optimization approach to enable VLMs to learn accurate spatial positions of visual objects, thereby enhancing multimodal reasoning capabilities.

Specifically, the proposed method, POLIA, requires VLMs to generate coordinate information for the visual evidence referenced in the reasoning chain. An additional Visual-Object-Level Intrinsic Advantage is then computed to supervise the learning of object coordinates. This intrinsic advantage is calculated based on the alignment between predicted and ground-truth bounding boxes of visual objects and is integrated with the traditional answer-level advantage.

Experimental results demonstrate the effectiveness of the proposed object-level intrinsic advantage, particularly on benchmarks that require precise visual reasoning.

**Compliance With Llm Reviewing Policy:**

Affirmed.

**Final Justification:**

The detailed responses have addressed my concerns, so I have decided to raise my score.

**Key Questions For Authors:**

Due to some unclear key details, I cannot give a high initial score. If the author can solve my concerns, especially Weaknesses 2, 3, and 4, I will improve my score.

**Limitations:**

yes

**Strengths And Weaknesses:**

### Strengths

- The proposed visual-object-level policy optimization is a meaningful approach for enhancing VLMs by encouraging accurate identification of visual evidence. Moreover, the introduced visual object grouping and intrinsic advantage computation are lightweight, which facilitates practical integration with the existing GRPO framework.
- The experimental results show that POLIA significantly outperforms GRPO across multiple tasks. The ablation study further demonstrates the effectiveness of incorporating visual-object-level intrinsic advantages.



### Weaknesses

1. The figures in the paper contain relatively low information density, with many repeated or highly similar visual elements, especially Figure 1.
2. Some key explanations and formulas are missing or insufficiently described:
    - How is the format validity term $R_{fmt}(c_i)$ computed?
    - What is the predefined matching rule for predicted boxes and ground-truth boxes? Additionally, how does POLIA handle cases where the model predicts a bounding box that does not correspond to any ground-truth box?
    - The formula for computing the object confidence score $s_{i,o}$ is not provided.
3. The training stage relies on ground-truth bounding boxes. It is unclear how these annotations are obtained. Are such annotations already available in the training datasets used in the experiments?
4. The reported “Average Improvements” in Table 1 appear inconsistent with the description in Section 5.2. In addition, the computation of these improvements is unclear, as I was unable to reproduce them based on the reported results.



The following are some minor issues:

- There are several typos, like the answer shown in Figure 17.
- Is "Qwen2.5-VL + GRPO" equivalent to the POLIA framework without the visual-object-level intrinsic advantage component?

---

> ### Author Rebuttal · Authors · 2026-03-30
>
> Hi Reviewer Jtum,
>
> Thank you for your time and detailed review. We appreciate your positive assessment of our overall approach. Your feedback is very helpful for improving the clarity and reproducibility of the paper. We provide our responses to each of your questions below.
> ## Q2. Additional Clarifications on Method Details
> >**2.1 Format validity term:** It consists of two parts: 1.Structural constraint: Outputs must include structured tags (e.g.`<answer>`, `<think>`); rewards are given if the format is satisfied to ensure the reasoning chain is parsable and well-formed. 2.Coordinate constraint: Regular expressions are used to detect bounding boxes with valid formats; rewards are given if valid coordinates are found. Format validity term = Structural constraint reward + Coordinate constraint reward.
>
> >**2.2 Bounding Box Matching and Unmatched Case Handling:**
>
> Matching mechanism: For each predicted box, we compute IoU with ground-truth boxes in the sample and use the maximum IoU for matching (a standard alignment strategy). Notably, only predictions whose max IoU exceeds the threshold are considered reliably aligned.
>
> Handling unmatched predictions: Boxes that do not meet the threshold are filtered out and excluded from the visual object set construction, as their semantic alignment is unstable and their forced inclusion may introduce noise and interfere with advantage estimation.
>
> >**2.3 Object confidence score:** It combines an IoU similarity term (spatial overlap) and a normalized L1 distance term (coordinate deviation) via weighted fusion, then is clipped to [0,1] (Section A.1). Sensitivity analysis is provided: https://postimage.me/image/pdzi.
> ## Q3. Whether Annotations Are Provided Originally
> Thank you for this important question. Ground-truth annotations are **provided by the original public datasets**, which already include manual annotations; no additional annotation is required. We use the public datasets VSR and TallyQA for training. Each dataset consists of questions, answers, images, image sizes, and bounding box annotations in JSON format. All baselines in our paper use the same datasets with identical data and annotation settings, ensuring a fair comparison.
> ## Q4. Clarifications on Average Improvements
> >**TLDR:** Section 5.2 reports the improvement **over GRPO**, while the “average improvements” in Table 1 report the average improvement **over all baselines**; the two are computed **over different scopes**.
>
> >**Action:** Thank you for raising this question, we will clarify the computation of the “average improvements” more clearly in Section 5.2.
>
> In Table 1, “Average Improvements” is calculated by first averaging the performance of multiple baselines under the same model scale and dataset category with equal weighting, and then subtracting this average from the POLIA performance to obtain the improvement. **For example, on the VSR dataset with the 7B model: Average Improvement = POLIA − 1/3 × ("Qwen2.5-VL" + "Qwen2.5-VL+DAPO" + "Qwen2.5-VL+GRPO") = 81.3 − 1/3 × (41.3 + 60.7 + 59.0) = 27.6333, which is reported as 27.6 after rounding to one decimal place.** In addition, the “over GRPO” phrasing in Section 5.2 is used to highlight the performance gain of POLIA over a single baseline (GRPO), whereas Table 1 reports improvements averaged over a broader set of baselines, so the statistics differ in scope.
> ## Other Minor Concerns
> **Q1. Figure Suggestions**
>
> >**Action:** Thank you for the suggestions on the figures and we agree that there is room for improvement in information presentation. Figure 1 is intended to provide a high-level overview using a simplified and consistent style to help readers quickly grasp the overall framework. In the revision, we will improve the figures by enhancing module distinction, reducing redundancy, and increasing information density where appropriate, while maintaining readability to better illustrate the method structure.
>
> **Minor issue1**
>
> We sincerely thank you for pointing out the typo in Figure 17 of the Appendix. In the revision, we will correct this error and further thoroughly proofread the entire manuscript and all figures to avoid similar issues.
>
> **Minor issue2**
>
> After removing the visual-object-level intrinsic advantage, POLIA’s objective is similar to Qwen2.5-VL + GRPO, and the two can be considered equivalent in this regard.  One of the key innovations of this work is the visual-object-level intrinsic advantage, which enables credit assignment over visual evidence and allows the model to reason based on correct visual objects, leading to POLIA’s significant improvement over standard GRPO.
>
> >Thank you again for your insightful feedback. We sincerely hope that our responses adequately address your concerns. Please do not hesitate to let us know if any points remain unclear—we would be happy to provide further clarification. We would greatly appreciate your consideration of a score update. We look forward to your response.

---

> > ### Author Rebuttal · Reviewer_Jtum · 2026-04-02
> >
> > Thank you for the detailed responses, which have addressed my concerns. I will increase my score accordingly.
> >
> > I hope these comments can help the authors improve the paper quality.

---

> > > ### Author Response · Authors · 2026-04-06
> > >
> > > Thank you very much for your positive evaluation and for adjusting your score. We sincerely appreciate your time and effort in carefully reviewing our work. We are glad that our clarifications and additional explanations were helpful in addressing your concerns. Your feedback has been very valuable in improving the clarity and overall quality of the paper.
> > >
> > > We truly appreciate your support and recognition of our work.

---

### Official Review · Reviewer_jG4W · 2026-03-13

**Soundness:** 2
**Presentation:** 3
**Significance:** 3
**Originality:** 3
**Overall Recommendation:** 4
**Confidence:** 4

**Summary:**

This paper proposes POLIA, a group-based reinforcement learning method for multimodal reasoning that augments standard answer-level advantages with a visual-object-level intrinsic advantage. The method groups candidate answers by the set of visual objects they reference, computes answer-level extrinsic rewards from final-answer correctness and formatting, then derives object-level intrinsic advantages using confidence-weighted rewards within each visual-object group. The paper evaluates POLIA on seven multimodal reasoning benchmarks and reports improvements over open-source baselines such as GRPO, DAPO, and GRIT, along with ablations on the two advantage terms, an analysis of visual object group convergence, and a small computational overhead study.

**Compliance With Llm Reviewing Policy:**

Affirmed.

**Final Justification:**

The rebuttal resolved my concerns, and I think it's necessary to add the experiments during rebuttal to the revised version, making the paper more complete.

**Key Questions For Authors:**

See Weaknesses.

**Limitations:**

For limitations, see weaknesses.

**Strengths And Weaknesses:**

Strength:
1. The paper addresses a real issue in multimodal RL, namely that answer-level rewards are often too coarse to assign credit to the visual evidence actually used during reasoning. The proposed decomposition into answer-level and object-level signals is intuitive.
2. Experimental results in Table 1 demonstrate that POLIA improves over the base Qwen2.5-VL-3B/7B model and over GRPO/DAPO on several benchmarks. The ablation curves in Figure 3 also support the claim that both the extrinsic and intrinsic terms contribute.
3. The computational overhead analysis in Figure 6 is useful. If the implementation is as described, the added grouping and intrinsic-advantage computation are indeed negligible relative to rollout cost, which is an important practical point for RL-based MLLM training.

Weaknesses:
1. The convergence analysis in Figures 4 and 5 shows that the number of distinct visual object groups decreases during training, but that is not the same as demonstrating that the model is assigning credit correctly to the right objects. A reduction in group diversity could also reflect mode collapse or prompt-format regularization rather than improved grounding. It's better to introduce a more direct validation of the claimed credit assignment mechanism.
2. The authors do not appear to demonstrate how to mitigate reward hacking, and there is no experiment specifically addressing this point. It's more like a plausible motivation than a demonstrated result.
3. It's better for the authors to compare some more recent and relevant baselines, especially for the 7B models.
4. The paper should also analyze the sensitivity to the intrinsic weight ($\omega$), the group size (N), the confidence score, et al.
5. Regarding presentation, the author did not correctly label the equations in this paper.

If the authors can address my concerns, I'd like to raise my score.

---

> ### Author Rebuttal · Authors · 2026-03-30
>
> Hi Reviewer jG4W,
>
> Thank you for your detailed review and recognition of POLIA. Your questions and suggestions are highly valuable. We provide new experiments and analyses below.
> ## Q1. Whether Group Convergence Caused by Mode Collapse
> >**TLDR:** Group convergence aligns with performance gains, variance reduction and ablation results, indicating the model learns more stable and effective visual evidence selection **rather than factors like mode collapse**.
>
> >**Action:** We agree that adding more direct credit assignment evaluation would strengthen the conclusions, and will add a key object masking experiment in Section 5.4 to validate this mechanism.
>
> **Correlation with performance:** As group number decreases, model accuracy increases (Figs. 4/5), showing a stable positive correlation. This contradicts mode collapse and prompt-format regularization, which usually degrade performance, and instead suggests the model increasingly focuses on effective visual evidence.
>
> **Effect of intrinsic advantage:** As noted at the end of Section 5.4, removing object-level intrinsic advantage clearly increases group numbers. This shows group convergence is not due to generic regularization, but is driven by the intrinsic advantage signal.
>
> Overall, the observations in Fig4/5 are not isolated; these results above support the effectiveness of the proposed object-level credit assignment mechanism.
> ## Q2. Whether POLIA Mitigates Reward Hacking
> >**TLDR:** POLIA aligns advantage signals with object-level visual evidence, preventing reward gains caused by guessing and **effectively mitigating reward hacking**.
>
> >**Action:** We acknowledge the importance of this issue in RL and multimodal reasoning, and will expand the discussion in Section 5.3.
>
> Prior methods rely on coarse rewards, leading to imprecise credit assignment and potential reward hacking. In contrast, POLIA introduces object-level intrinsic advantages as fine-grained reward signals, enabling more precise credit assignment and reducing reward hacking. Ablation in Section 5.3 further supports this claim, as removing intrinsic advantages leads to a significant performance drop, indicating that coarse rewards provide weaker grounding and are more prone to imprecise attribution and spurious correlations.
> ## Q3. Latest Baseline Comparisons
> >**TLDR:** We report answer accuracy of Qwen3-VL-8B/4B, as well as POLIA built upon these two backbones, on VSR and TallyQA, using the same metrics as Tab1 of the paper. **POLIA consistently outperforms the latest baseline.**
>
> >**Action:** Thank you for your suggestions, we will include the experimental results in Section 5.2.
>
> Hardware: NVIDIA A800-80GB (NVLink); 8B uses 3 cards, 4B uses 2 cards. Hyperparameters: All settings follow Section A.1, no extra tuning. Dataset & evaluation: VSR and TallyQA, same splits and GPT‑4o judge scoring as Table 1.
> |Model|VSR|TallyQA|
> |-|-|-|
> |Qwen3-VL-8B|68.5|55.9|
> |POLIA-8B|**75.2**|**77.0**|
> |**Improvements**|**6.7**|**21.1**|
> |Qwen3-VL-4B|65.9|54.4|
> |POLIA-4B|**74.7**|**61.6**|
> |**Improvements**|**8.8**|**7.2**|
>
> ## Q4. Sensitivity Analysis
> >**TLDR:** We add VSR (POLIA-3B) results on intrinsic weight and confidence score. Together with group size in the paper, all hyperparameters show smooth trends over wide ranges, indicating **low sensitivity**, with chosen values in stable, near-optimal regions.
>
> >**Action:** We appreciate your insightful comments, we will include the sensitivity analysis in the Appendix.
>
> **Intrinsic weight:** Performance improves, then plateaus with slight decline; ω=1.0 lies in this stable region. This shows intrinsic advantage improves performance, is robust over a wide ω range, and overly large weights may weaken the extrinsic signal. Curve plots: https://postimage.me/image/pdhY
> |Intrinsic Weight|Answer Accuracy(%)|
> |-|-|
> |0.0|53.5|
> |0.5|65.1|
> |1.0|71.9|
> |1.5|67.4|
> |2.0|67.8|
>
> **Confidence score:** Performance rises with IoU weight, then stabilizes and declines; the chosen setting is near-optimal. IoU dominates spatial alignment, L1 complements position, and combining both outperforms either alone with stable results across a wide range. Curve plots: https://postimage.me/image/pdzi
> |IoU / L1 Weight|Answer Accuracy(%)|
> |-|-|
> |0.0/1.0|51.9|
> |0.3/0.7|63.0|
> |0.5/0.5|68.4|
> |0.7/0.3|71.9|
> |1.0/0.0|60.8|
>
> **Group size:** N = 8 underperforms 16 and 24, while the latter are similar and stable, indicating limited exploration harms performance and gains saturate once diversity is sufficient (Fig4/5, Sec5.4).
>
> ## Q5.Equation Formatting
> >**Action:** Thank you for pointing out this issue. We will **number all equations** uniformly in the revision to improve the paper’s consistency and readability.
> ##
> >Thank you again for your valuable feedback. We hope our responses address your concerns. If so, we would appreciate your consideration in updating scores. We look forward to your reply.

---

> > ### Author Rebuttal · Reviewer_jG4W · 2026-04-04
> >
> > Thanks for the rebuttal, I look forward to seeing all the details in the paper, I will keep my score.

---

> > > ### Author Response · Authors · 2026-04-06
> > >
> > > Thank you for the thoughtful and constructive feedback. We are grateful for your acknowledgment that the main concerns have been addressed. Regarding the additional details, we will incorporate these clarifications into the revision, including more experiments, analyses, and improved presentation to make the work more complete and transparent.
> > >
> > > We sincerely appreciate your valuable feedback and the time and effort you have invested in reviewing our work. Your comments have been extremely helpful in improving the clarity and completeness of the paper, and we truly benefit from your suggestions. Since the key concerns have now been resolved, we hope this clearer and more complete version of the work may support a more positive overall assessment. Thank you again for your time.

---

### Decision · Program_Chairs · 2026-04-30

**Decision:**

Accept (regular)

**Comment:**

This paper proposes POLIA, a group-based RL method for multimodal reasoning that augments standard answer-level advantages with visual-object-level intrinsic advantages for finer-grained credit assignment. Reviewers found the motivation clear, the method well designed, and the empirical gains across multimodal reasoning benchmarks strong.

The main concerns were about validation of the object-level credit assignment mechanism, sensitivity to key hyperparameters, applicability beyond the original backbones, and clarity of several implementation details. The rebuttal addressed these well, with additional experiments on newer Qwen3-VL models, sensitivity analyses, further discussion of singleton groups and annotation dependence, and clearer methodological explanations. Reviewers who raised these concerns indicated they were resolved, including score increases.

Overall, this is a technically solid and practically meaningful contribution with strong empirical support.